# Recent Trends in Foliar Nanofertilizers: A Review

**DOI:** 10.3390/nano13212906

**Published:** 2023-11-06

**Authors:** Yanru Ding, Weichen Zhao, Guikai Zhu, Quanlong Wang, Peng Zhang, Yukui Rui

**Affiliations:** 1College of Resources and Environmental Sciences, China Agricultural University, Beijing 100193, China; dingyr314@163.com (Y.D.); weichenz@cau.edu.cn (W.Z.); zgk18451176676@163.com (G.Z.); wql18764927102@163.com (Q.W.); 2Department of Environmental Science and Engineering, University of Science and Technology of China, Hefei 230026, China

**Keywords:** nanofertilizers, foliar fertilization, heavy metal stress, salt stress, drought stress

## Abstract

It is estimated that 40–70%, 80–90% and 50–90% of the conventional macronutrients N, P and K applied to the soil are lost, respectively, resulting in considerable loss of resources. Compared to conventional fertilizers, nanofertilizers have the advantages of controlled release, high nutrient utilization, low cost and relatively low environmental pollution due to their small size (1–100 nm) and high specific surface area. The application of nanofertilizers is an up-and-coming field of agricultural research and is an attractive and economical substitute for common fertilizers which can boost global food productivity sustainably. Foliar fertilization is a popular way to satisfy the needs of higher plants. Because of its small application dose, faster nutrient uptake than soil application and relatively less environmental pollution, foliar fertilization is more popular among plants. It can be seen that nanofertilizers and foliar fertilization are the hotspots of attention at present and that current research on the foliar application of nanofertilizers is not as extensive as that on soil application. Based on this background, this paper provides an overview of various applications of foliar spraying of nanofertilizers in agriculture, including applications in improving crop yield and quality as well as mitigating heavy metal stress, salt stress and drought stress.

## 1. Introduction

Fertilizers are synthetic or natural substances mixed into the soil or sprayed on the foliage of plants to deliver one or more nutrients to foster crop nutrition [1,2]. Although synthetic fertilizers can improve crop growth, they are too inefficient to meet current agricultural needs [3]. Moreover, excessive use of synthetic fertilizers increases production costs and environmental risks, which is detrimental to sustainable agricultural development [3]. The use of nanofertilizers allows for better nutrition management, increasing crop productivity and providing support for plant development under environmental stresses [4,5] (Figure 1). Compared with traditional fertilizers, nanofertilizers are used in much lower amounts and can reduce environmental pollution, eutrophication and groundwater contamination to a great extent [6]. In addition, nanofertilizers protect nutrients from leakage or volatilization and therefore retain their fertility longer than traditional fertilizers [3]. Nanofertilizers can enhance the efficiency of nutrient delivery through targeting and gradual release of nutrients, as well as reduction in fertilizer application rates [7]. A study of nanofertilizers versus their conventional analogs by Kah et al. showed that nanofertilizers for micronutrients improved by 18% and nanofertilizers for macronutrients improved by 29% compared to regular fertilizers [8]. In addition, with sizes ranging from 1–100 nm, nanofertilizers are more easily absorbed by plants, thus increasing the physiological efficiency and general yield of the crop [9]. Generally speaking, nanofertilizers hold great promise for plant nutrition as they can increase crop yields, minimize the risk of environmental pollution and lower application costs.

Depending on the way in which nutrients are absorbed by the crop, options for nano-fertilization can be divided into seed priming, soil application and foliar application. Nanoparticles undergo a series of reactions in the soil which cause changes in physicochemical properties, thus affecting the absorption of nanoparticles by plants [10]; for example, excess phosphorus is chemically bonded to other elements in the soil and is not absorbed by plants [11]. Also, root exudates can affect the surface chemical properties of nanoparticles and are a barrier to root uptake of nanoparticles [12,13]. However, seed priming and foliar fertilization will improve those problems. Seed priming is an empirical practice and pre-experimentation is required for different seed batches. Seed nano-priming technology refers to the use of nanoparticles as seed pretreatment agents to improve seed germination, growth and resistance, which is a relatively new research direction. Seed initiation with nanofertilizers prior to planting reduces unnecessary losses and environmental impacts; however, the initiation formulation and concentration need to be carefully selected to avoid adverse effects [14]. Work needs to be done to understand how NPs act as seed pretreatment agents since seed priming is an emerging field, and more research is needed to develop a complete understanding of NP delivery systems. Since this article is mainly an overview of foliar fertilization, it will not discuss seed priming as a fertilization approach in depth. Foliar fertilization has a more direct and targeted response than soil fertilization, the amount of fertilizer added to the soil is low and nutrients can be properly used to reduce environmental pollution [15]. Foliar application delivers nutrients directly to the target organ, contributing to the mitigation of the negative effects of stress [16]. In addition, it has been shown that foliar nanofertilizers are friendly for use in the field as they can be applied gradually to plants at a much more controlled rate, which reduces the symptoms of toxicity that may follow soil application [8]. Moreover, foliar application is simpler and cheaper than adding nanofertilizers to the soil [17]. There is increasing recognition that sustainable plant management through foliar fertilization can tackle problems like high losses of fertilizers from soil application and restrictions on nutrient delivery to plant organs due to environmental conditions [18].

Based on this background, this paper briefly introduces foliar fertilization and provides an overview of the current status of foliar nanofertilizer application and development, mainly in the areas of crop yield enhancement and stress mitigation.

## 2. Pathway of Nanofertilizers into Plants

Essential nutrients are most commonly used in soil and plant leaves, and soil fertilization is more popular and more effective for the larger nutrient requirements [19]. However, in certain cases, foliar fertilization has become a widespread and common approach to crop management due to its characteristics of being more cost-effective and efficient [19].

The leaves of a plant can protect the plant from water loss, pests and pathogens while permitting the exchange of gases for photosynthetic reactions [20]. The surface of the leaf usually consists of features: trichomes, stomata, and phloem pores. NPs are absorbed by the foliage in two ways, i.e., the cuticle pathway and the stomatal pathway [21] (Figure 2). NPs with a diameter of less than 4.8 nm can pass through the cuticular channel directly into the leaf [22], while NPs with larger particle sizes can enter via stomata. Due to the high density of the stomata themselves, the stomatal pathway is considered to be a more efficient way of uptake of NPs [23]. After NFs are taken up by the stomata, they are transferred to the rest of the plant via the phloem. Nanoparticles can enter the phloem in two ways, either directly from the phloem cells into the phloem or through the interstices of the phloem cells into the phloem [24]. Nanofertilizers can promote rapid access to nutrients at plant growth sites, thereby increasing chlorophyll production, photosynthetic rate, and eventually growth and development of plants [25].

The composition of the stratum corneum and the surface properties of NPs may influence the efficiency of NP uptake by the stratum corneum, as shown by Avellan et al. [26]. For example, stomatal size affects the entry of NPs, and NPs may affect stomatal size; the adherence of NPs to leaves affects the effective delivery to plants [27]; and the size of sieve plate pores can limit the effective transportation of particles in plants [24] (Figure 2).

## 3. Agricultural Application of Foliar Nanofertilizers

Nanofertilizers are becoming increasingly important in improving nutrient utilization efficiency owing to their unique properties [28]. Nanofertilizers help to release nutrients in a slow, controlled way in order to deliver nutrients to the target location, causing minimal losses [29]. Nanofertilizers provide greater uptake and retention than conventional fertilizers owing to their tiny size [30]. Nanofertilizers can improve the physiological and biochemical indicators of plants, such as photosynthetic rate and absorption efficiency of nutrients, and boost the defense system of the plant [7]. Several studies have reported that zinc NFs have better physicochemical properties and act positively in promoting seed sprouting and plant growth [31,32]. Ahmed et al. showed that sulfur nanofertilizers not only mitigate arsenic toxicity but also improve rice yield and quality [33]. Hu et al. demonstrated that low levels of TiO_2_ NPs improved plant nutritional quality without causing significant oxidative stress [34]. Overall, nanofertilizers are more effective than traditional fertilizers and have good prospects for development.

Foliar fertilization is more efficient than soil fertilization and is a useful approach to satisfying the requirements of higher species [35] (Figure 3). Soil application is more environmentally hazardous and nanofertilizers are less bioavailable in the soil versus foliar application, which has lower environmental risks, so foliar application of nanofertilizers is usually preferred by plants [36]. Foliar-applied fertilizers have a low exposure dose, can be applied repeatedly and can be time-applied according to the weather to avoid nutrient losses [11]. Furthermore, foliar application provides faster nutrient uptake than soil application [37]. Figure 3 illustrates the advantages of foliar fertilization and its positive effects on plants.

The next part of the paper is an analysis of reports from the past five years, which concludes that foliar application of nanofertilizers can improve crop yield and quality as well as mitigate the adverse impacts of environmental stresses.

### 3.1. Improvement in Crop Yield and Quality

Foliar application of nanofertilizers can enhance the utilization efficiency of fertilizers as well as crop production and quality, and to a certain extent reduce adverse effects (Table 1).

Zinc (Zn), as an essential micronutrient, has a significant influence on plants, including the synthesis of proteins, DNA and RNA, as well as is a cofactor necessary for numerous antioxidant enzymes. There have been many studies on the favorable effects of foliar spraying of zinc-based nanofertilizers on crops. Lorenzo et al. found that owing to the improved ability of ZnO NPs to penetrate the leaves, ZnO NPs improved the growth and physiology of coffee and more positively affected fruit and quality than ZnSO_4_ [18]. Garcia-Lopez et al. demonstrated that foliar sprays of ZnO NPs (1000 mg/L and 2000 mg/L) increased the antioxidant capacity of *Habanero pepper* fruits and also significantly improved fruit quality [38]. Kolencik et al. found that both TiO_2_ and ZnO NPs treatment enhanced the yield and nutrient parameters of sunflower, but the TiO_2_ NPs treatment was potentially toxic, whereas no toxicity by ZnO NPs treatment was detected [39]. Davarpanah et al. concluded that foliar spraying of B or Zn nanofertilizers at lower concentrations could boost the production of pomegranate, and the characteristics of the fruits were unaffected [40]. There are also other studies showing that foliar application of zinc nanofertilizers not only increased leaf number and essential oil levels but also significantly increased plant growth, yield and nutrient content [41,42,43,44,45,46]. It can be seen that foliar application of Zn nanofertilizers positively affects the improvement of crop yield, quality, nutrient quantity and physiological parameters with probably no potential toxicity, but it should be mentioned that the difference in optimal concentration may be large for different plants.

Similarly, there are also many studies suggesting that foliar application of other nanofertilizers can promote plant yield and quality. Foliar application of chitosan-silica nanofertilizers improved antioxidant defense capacity and boosted the chlorophyll content of maize plants, leading to improved growth and yield [47]. Foliar application of MgO NPs significantly affected chlorophyll content and plant yield and also improved the height and leaf number of cotton plants [48]. Foliar spraying of Fe NPs was beneficial in improving nutrient content, flowering, and fruit yield and quality [49,50]. Foliar application of Ag NPs at 15 mL/L was most effective in boosting stem thickness, leaf area, photosynthetic efficiency, flowering rate and fruit physicochemical properties of peach when compared to other groups [44]. Foliar application of MnO NPs has a significant effect on pumpkin yield [29]. Foliar application of Ca NPs at 0.50 g/L was the most effective in reducing fruit cracking as compared to other groups [51]. Thus, it can be deduced that foliar application of nanofertilizers at certain concentrations has been found to have positive effects on plant quality, chlorophyll content and yield, but particular attention needs to be paid to the application concentration.

**Table 1 nanomaterials-13-02906-t001:** Application of nanofertilizers in enhancing crop yield and quality.

Nanofertilizers	Crops	Impacts	Reference
ZnO NPs	*Coffea arabica* L.	Improvement in coffee fruit quality, positive effects on growth and physiology, and a 55% increase in net photosynthetic rate.	[18]
ZnO NPs	*Habanero pepper*	Increase in the antioxidant capacity and improvement in fruit quality.	[38]
ZnO NPs and TiO_2_ NPs	Sunflower	Increase in oil production and a better quantitative and nutritional effect.	[39]
Nanofertilizers of Zn and B	Pomegranate (*Punica granatum* cv. Ardestani)	Low amounts of B or Zn nanofertilizers can increase fruit yields.	[40]
ZnO NPs	Foxtail Millet (*Setaria italica* L.)	Increase in photosynthetic efficiency, transpiration, and enzyme activities.	[41]
ZnO NPs	Turmeric (*Curcuma longa*) plant	Increase in growth, yield, nutritional quality, and biochemical indicators.	[42]
Zn and B	*Litsea cubeba*	Improvement in essential oil.	[43]
Zn NPs and Ag NPs	Peach	Increase in stem thickness, total chlorophyll, flowering rate and yield.	[44]
ZnO NPs	Faucet plants	Increase in biomass and essential oil production.	[46]
Chitosan-silica nanofertilizers	Maize plants	Improvement in yield, antioxidant defense capacity, and photosynthetic efficiency.	[47]
MgO NPs	Cotton plants	Increase in chlorophyll content, plant yield, plant height, and leaf number.	[48]
Fe NPs	Washington navel orange trees	Enhancement of nutrient content, flowering rate, and fruit yield and quality.	[49]
Fe NPs	Broad bean grown	Enhancement of plant growth, pod yield, and quality.	[50]
ZnO NPs	Squash plants	Enhancement of plant growth and yield.	[29]
Nano-calcium fertilizer	Pomegranate (*Punicagranatum* cv. Ardestani)	Reduction in fruit cracking.	[51]

### 3.2. Mitigation of Environmental Stress

Environmental stress effects can alter ecosystem processes [52], disrupting ecosystem equilibrium, which in turn disrupts the balance of environments associated with food production, potentially leading to reduced crop yields [53]. Heavy metals, salinization, drought and high temperatures are all key environmental stressors that have serious impacts on the productivity and quality of crops worldwide [54]. A variety of strategies have been sought to improve the capacity of plants to withstand these numerous environmental stresses [55]. Owing to their high efficiency and slow release, nanofertilizers have become a suitable option for mitigating environmental stress effects [56,57] and enhancing crop cultivation in adverse environments [58,59]. Several studies have confirmed the positive effects of NPs on plants under temperature stress, including improvements in photosynthetic capacity [60] and promotion of growth and development [61]. However, not enough research has been conducted on heat stress; thus, this article describes the application of nanofertilizers in mitigating heavy metal stress, salt stress and drought stress, which are aspects of more concern in current research (Table 2).

#### 3.2.1. Heavy Metal Stress

Heavy metals are taken up by plants and accumulate in grain crops for human and animal consumption, seriously endangering crop growth and human health [62,63]. Many investigations have suggested that nanoparticles can mitigate heavy metal stress on plants [64,65]. Foliar application of Se and Si NPs alleviates metal stress in rice and improves brown rice yield and quality [66]. Foliar application of ZnO NPs mitigated Cd contamination and increased plant height and biomass as well as chlorophyll concentration of maize plants [67]. Foliar application of TiO_2_ NPs significantly reduced stem Cd content, which contributed significantly to the reduction in Cd-induced toxicity; however, soil application of TiO_2_ NPs increased the absorption of Cd by maize in Cd-contaminated soil [68]. From the above, it is evident that the foliar application of nanofertilizers has a mitigating effect on soil heavy metal contamination and may be more useful than soil application to a certain extent. However, it is to be noted that the presence of heavy metals may promote the uptake and enrichment of nanoparticles in plants with the emergence of co-toxicity, which leads to food safety issues [69].

#### 3.2.2. Salt Stress

Salinity is recognized to be among the major abiotic stresses limiting crop yield worldwide [70,71]. Salinity stress limits growth, reduces biomass, leads to chlorophyll degradation and alters water status [72]. More eco-friendly mitigation strategies for salt stress to enhance crop yields are critical for the agricultural sector [73]. Abdelaal et al. demonstrated that foliar application of silicon can mitigate the adverse influence of salt stress on *sweet peppers* by improving water status, boosting photosynthetic rate, modulating certain osmolytes and phytohormones, and improving antioxidant enzyme activities [74]. Hajihashemi et al. demonstrated that foliar application of silica nanofertilizers can significantly increase the salt resistance of wheat plants by improving enzymatic and nonenzymatic antioxidant systems [75]. Perez-Labrada et al. suggest that foliar application of Cu nanoparticles enhanced salt tolerance by improving the Na+/K+ ratio as well as stimulating antioxidant mechanisms in plants [76]. Sheikhalipour et al. proved that foliar application of Cs-Se NPs promoted plant production by enhancing the content of leaf photosynthetic pigments and also mitigated oxidative damage under salt stress conditions by raising the activities of SOD, POD and CAT enzymes [77]. Mustafa et al. revealed that foliar application of low doses of TiO_2_ nanoparticles improved germination characteristics and water and osmotic potentials of wheat and helped to increase plant tolerance to salt stress [78]. Moreover, a review showed that the application of Zn and ZnO NPs alleviated the adverse effects of salt stress on crop yield and quality as well as increased protein content and antioxidant capacity [37]. Silicon nanofertilizers possess positive effects in alleviating salt stress [54,79,80]. For example, the results by Alsaeedi et al. showed that amorphous silica nanoparticles (Si NPs) contributed to the normal growth of cucumber plants under salt stress without any noticeable water deficit symptoms throughout the growing season [81]. However, foliar application of silicon fertilizers has not been fully studied and is a novel direction of research. Si nanoparticles as coatings sprayed with other nanoparticles are also relatively new. A study has shown that foliar application of ZnO NPs and ZnO-Si NPs had different effects on pea plants under salt stress [82]. Higher concentrations of ZnO NPs produced some phytotoxic effects, whereas ZnO-Si NPs had no toxicity to the plant under physiological conditions and even had a slight stimulatory effect at higher concentrations. The foliar application of nanofertilizers is one of the trends in alleviating salt stress.

#### 3.2.3. Drought Stress

Drought conditions are also a key constraint on crop yield, causing plants to experience many unfavorable stresses at the morphological, physiological and molecular levels [83] and affecting plant growth, physiology and yield [84]. Drought stress is usually the result of drought-induced stomatal closure leading to oxidative stress, in turn resulting in an increase in ROS production in chloroplasts and mitochondria [85,86]. Drought stress limits the photosynthetic process by altering the content of chlorophyll and other photosynthetic pigments, thus leading to the cessation of plant growth [87,88,89]. Moreover, as drought gets worse, soil salinization and calcification increase, which in turn leads to a significant decrease in productivity [90]. Foliar application of nanofertilizers may be the best option for improving yields in the water-scarce semi-arid tropics, as a large amount of water is required for their solubilization and absorption by the root system [91]. Foliar application of nanofertilizers as growth regulators promote crop development and productivity under drought conditions [91]. For example, foliar application of ZnO NPs improved the yield and crop quality [41,92] as well as the nutritional quality of seeds [93] and improved stomatal conductance and the crop drought stress index [93]. Furthermore, Moitazedi et al. showed that foliar application of Zn fertilizer greatly remediated the impact of drought stress on the membrane stability index (MSI) [94]. Studies have also shown that in conditions of water deficit, foliar spraying of Fe and Zn nanofertilizers improved physiological properties and seed yield of beans with normal irrigation [84,95]. Foliar application of K-nano chelate provides an improvement in growth as well as physiological and biochemical characteristics, increases quantitative and qualitative traits and mitigates the negative impacts of water stress [96]. Foliar application of Mg nanofertilizers and chitosan fertilizer increased total chlorophyll yield, seed yield and oil content, as well as alleviating drought stress [97]. In addition to several metallic nanofertilizers, non-metallic nanofertilizers have also been used in many applications. For example, Sharf-Eldin et al. found that plants treated with foliar application of SiO_2_ NPs showed a significant increase in growth index despite drought stress [98]. Furthermore, a review found that TiO_2_ NPs applied at low concentrations through leaves improved crop performance by increasing the yield and quality of crops under drought stress [99]. Foliar application of both curcumin nanoparticles and ammonium glycyrrhizinate nanoparticles can mitigate the unfavorable impacts of drought on the development of soybeans [100].

In conclusion, foliar application of nanofertilizers under abiotic stress can improve plant enzyme activities and enhance plant antioxidant capacity. These improvements can improve crop resilience to stress and thus improve yield and quality.

**Table 2 nanomaterials-13-02906-t002:** Application of nanofertilizers in mitigation of environmental stress.

Types of Nanofertilizers Elements	Type of Environmental Stress	References
Se nanofertilizers (Se NPs, Cs-Se NPs)	Mitigation of metal stress.	[66,77]
Zn nanofertilizers (Zn and ZnO NPs)	Mitigation of metal stress, salt stress, and drought stress.	[37,41,67,84,92,93,94,95]
Si nanofertilizers (Si NPs and SiO_2_ NPs)	Mitigation of metal stress, salt stress, and drought stress.	[66,74,75,98]
Fe nanofertilizers (Fe_2_O_3_ NPs)	Mitigation of salt stress and drought stress.	[84,101]
Ti nanofertilizers (TiO_2_ NPs)	Mitigation of metal stress, salt stress, and drought stress.	[68,78,99]
K nanofertilizers (KO_2_ NPs)	Mitigation of drought stress.	[96]
Mg nanofertilizers (MgO NPs)	Mitigation of drought stress.	[97]
Cu nanofertilizers (Cu NPs)	Mitigation of salt stress.	[76]

## 4. Conclusions and Future Perspectives

As can be seen from publications in recent years, the foliar application of nanofertilizers has attracted widespread attention and can be applied alone or as a supplement to soil fertilization. Compared to soil application, foliar application of fertilizers acts directly on the crop, is more accessible and is not transformed by reaction with other substances in the soil. Foliar sprays are applied in smaller amounts, making them more economical and less harmful to the environment. In general, foliar nanofertilizers represent an excellent option for sustainable agriculture and have great potential for application. At the same time, however, there is a distance to be covered before large quantities of foliar nanofertilizers can be put into practical application, as the toxicity and environmental residue issues of nanoparticles remain unresolved and the optimal action concentration of nanofertilizers remains to be determined in the near future. The following is a list of current issues and expectations for the future:

(1) The optimal application concentration varies from crop to crop, and comprehensive trials are needed to determine the optimal application concentration and application time to avoid environmental pollution and potential phytotoxicity.

(2) Deeper studies are necessary on the action mechanism of fertilizers after they enter the foliage, to better determine the specific site of action of nanoparticles.

(3) Comparative studies on different application methods, such as seed irrigation, seed priming, nutrient solution root dipping, foliar spraying, etc., are needed to determine the most suitable application methods under different conditions.

(4) There is a need to focus on different nanomaterials and fertilization methods; the same nanomaterials may have different results under different application methods.

(5) There is still a long way to go before large-scale implementation, so more focus should be placed on the toxicity issue, the impact on human health and economic benefits.

## Figures and Tables

**Figure 1 nanomaterials-13-02906-f001:**
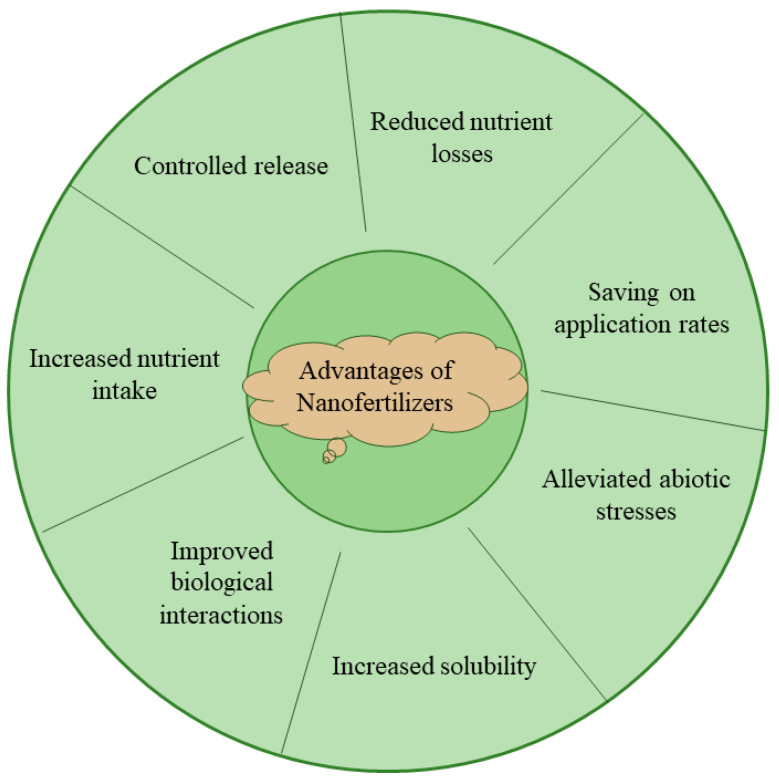
Advantages of nanofertilizers over traditional fertilizers.

**Figure 2 nanomaterials-13-02906-f002:**
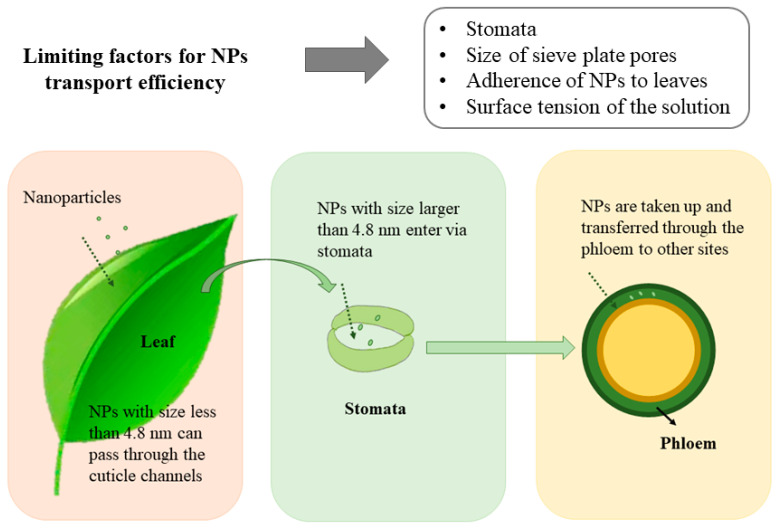
Pathways into plants for foliar application of nanoparticles and influencing factors.

**Figure 3 nanomaterials-13-02906-f003:**
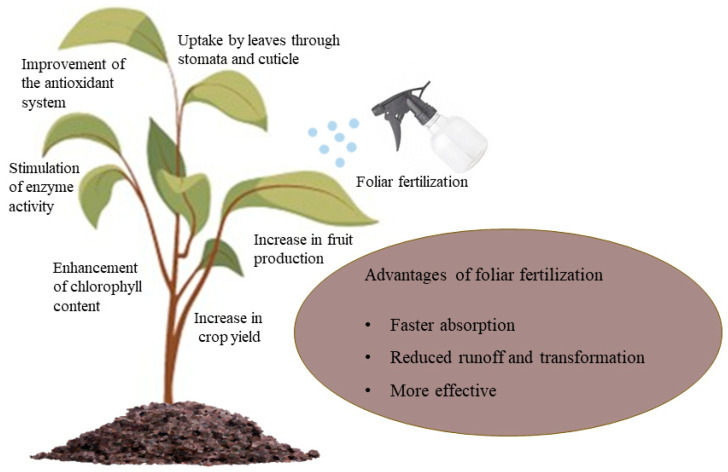
Foliar application.

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
