# Peer review of "Recent Trends in Foliar Nanofertilizers: A Review"

_nanomaterials, 2023, doi:10.3390/nano13212906_

Round 1

Reviewer 1 Report

Comments and Suggestions for Authors

The manuscript is of wrathful consideration for plant scientists and agronomists. The write-up of whole manuscript is up to mark. English quality is acceptable but minor revision needed. However, there are some suggestions that need to be incorporated before final acceptance as elaborated below:

-          The abstract should contain some quantitative results/findings.

-          Please note the opening paragraph of the introduction could provide stronger context to the paper, and, similarly, the findings at the end could potentially be richer.

-          All manuscripts lack of line numbers which is too difficult for the reviewer to mention the line number for corrections. In the abstract section, the Author should explain briefly why this study is required to be conducted. 

-          Results presentations should be revised profoundly. 

-          I suggest that the author should provide more justification for your study (specifically, you should expand upon the knowledge gap in the abstract, introduction, and all other sections being filled) which should be improved upon before Acceptance. 

-          The discussion need to revised and need to make it more focused based on results.

-          English quality needs moderate revision

Comments on the Quality of English Language

Moderate English revision 

Reviewer 2 Report

Comments and Suggestions for Authors

Round 2

Reviewer 1 Report

Comments and Suggestions for Authors

Accept

Reviewer 2 Report

Comments and Suggestions for Authors

The authors have addressed all my comments.